# Private and Communication-Efficient Algorithms for Entropy Estimation

**Gecia Bravo-Hermsdorff**
Department of Statistics
University College London
gecia.bravo@gmail.com

**Róbert Busa-Fekete**
Google Research
busarobi@google.com

**Mohammad Ghavamzadeh**
Google Research
ghavamza@google.com

**Andres Muñoz Medina**
Google Research
ammedina@google.com

**Umar Syed**
Google Research
usyed@google.com

## Abstract

Modern statistical estimation is often performed in a distributed setting where each sample belongs to a single user who shares their data with a central server. Users are typically concerned with preserving the privacy of their samples, and also with minimizing the amount of data they must transmit to the server. We give improved private and communication-efficient algorithms for estimating several popular measures of the entropy of a distribution. All of our algorithms have constant communication cost and satisfy local differential privacy. For a joint distribution over many variables whose conditional independence is given by a tree, we describe algorithms for estimating Shannon entropy that require a number of samples that is linear in the number of variables, compared to the quadratic sample complexity of prior work. We also describe an algorithm for estimating Gini entropy whose sample complexity has no dependence on the support size of the distribution and can be implemented using a single round of concurrent communication between the users and the server. In contrast, the previously best-known algorithm has high communication cost and requires the server to facilitate interaction between the users. Finally, we describe an algorithm for estimating collision entropy that matches the space and sample complexity of the best known algorithm but generalizes it to the private and communication-efficient setting.

## 1 Introduction

Statistical estimation has traditionally focused on minimizing the number of samples needed to estimate properties of a distribution. In the 'big data' era, statisticians and computer scientists have also tried to minimize the space complexity of estimation algorithms, particularly in the streaming setting. More recently, the increasing prevalence of mobile computing has led to a focus on the privacy and communication costs of statistical estimation. In this paper, we consider the following setting: a set of users each draw one sample from a distribution, and share information about their sample with a central server. The central server then uses the collected data to estimate a property of the distribution. Users are concerned with preserving the privacy of their sample, and also with minimizing the amount of data that is transmitted to the server.

For example, consider the problem of detecting fingerprinting on the web [31]. Many websites track users across the web without their consent by recording (enough) information about their devices (e.g., installed fonts, operating system, timezone, etc.), a practice known as "browser fingerprinting". Entropy is the standard metric used to quantify the identifiability of the collected fingerprints. So a

private and distributed method for estimating entropy can be used by a browser to warn users that this covert tracking could occur, without ever storing the fingerprints themselves.

The study of entropies has an extensive and rich history in mathematics and sciences. Related quantities called "entropy" appear in many contexts (thermodynamics, information theory, dynamical systems [35], category theory [9], etc.). These may be broadly thought as measures of information of a system or process obeying certain properties, which, in turn, lead to natural measures of disorder, randomness, outcome diversity, information content, uniformity, etc.

In this paper, we study private and communication-efficient algorithms for estimating certain entropies of a distribution. Specifically, we give algorithms for estimating the following entropies, which are widely-used in many scientific fields to quantify the uncertainty, diversity and spread of a discrete distribution:

- *Shannon entropy* [33], a fundamental quantity in information theory.
- *Gini entropy* (also known as Tsallis entropy [38] of order 2, or (one minus the) second frequency moment). Some of its applications include measuring ecological diversity [34, 27], market competition among firms [21], effective size of political parties [26], and suitability of features to split on during decision tree learning [30].
- *Collision entropy* (also known as Rényi entropy [32] of order 2). Some of its applications include measuring the quality of random number generators [36], and determining the number of reads needed to reconstruct a DNA sequence [28].

Our algorithms are implemented in either the *non-interactive* model (for the Gini and collision entropies), in which all users simultaneously exchange information with the server during a single round of communication, or the (stronger) *sequentially interactive* model (for the Shannon entropy), in which the server queries users one at a time, possibly in an adaptive manner [22]. When analyzing the communication complexity of an algorithm, we prove bounds on the number of bits that each user transmits to the server. However, the server is allowed to broadcast an arbitrary amount of information to the users (this is also called the *blackboard* model [16]), including shared random bits (also known as the *public coin* model [1, 2, 23]). When analyzing the privacy of our algorithms, we use the framework of *local differential privacy* [17], which ensures that the server learns very little about each user's data.

**Our contributions:**

- A sequentially interactive $\alpha$-local differentially private algorithm for estimating the Shannon entropy of a joint distribution on $d$ variables within $\epsilon d$ error using $\tilde{O}(d/\alpha^2\epsilon^3)$ samples and $O(1)$ bits per sample. Our analysis assumes that each of the $d$ variables has a constant support size and that their conditional independence graph is a tree. We also describe algorithms that have better dependence on $1/\epsilon$ in certain special cases, such as when the tree has low diameter or is a chain. Our algorithms achieve $O(1)$ communication complexity by observing only two or three of the $d$ variables in any single sample; we call these *pair* and *triplet observations*. The only previously known algorithm for estimating the Shannon entropy of a tree-structured distribution from *pair* observations is a non-interactive algorithm due to Chow and Liu [12]. We prove that any non-interactive algorithm requires $\Omega(d^2)$ *pair* observations to achieve $O(d)$ error. We also prove that, for any sequentially interactive algorithm, $\Omega(d/\epsilon)$ pair observations are necessary to achieve $O(\epsilon d)$ error.

- A non-interactive $\alpha$-local differentially private algorithm for estimating the Gini entropy of a distribution within $\epsilon$ error using $O(b^2 \max\{1 - g, 2^{-b}\}/(\alpha^2\epsilon^2))$ samples, $b$ bits per sample, and $\tilde{O}(b)$ space, where $g \in [0, 1]$ is the Gini entropy of the distribution. The best previous algorithms [10] either have a sample complexity that depends on the support size $k$ of the distribution, or are sequentially interactive, and also require $\Omega(k)$ bits per sample and $\Omega(k)$ space. Also our error bound holds with high probability instead of only in expectation.

- A non-interactive $\alpha$-local differentially private algorithm for estimating the collision entropy of a distribution with support size $k$ within $\epsilon$ error using $\tilde{O}(b^2k^2/(\alpha^2\epsilon^2 \min\{k, 2^b\}))$ samples, $b$ bits per sample, and $\tilde{O}(b)$ space. Setting $b = \log k$ and $\alpha = O(1)$ recovers the sample and space complexity guarantees of the non-interactive algorithm from [36] up to logarithmic factors, and thus, our algorithm generalizes the previously best known algorithm to the private and communication-efficient setting.

## 2 Related Work

There is a very extensive literature on distributed statistical estimation under communication constraints (see [40] for the paper that appears to have started this thread). Variations on the problem include whether communication is allowed between users, whether communication happens in one or multiple rounds, whether there is a shared source of randomness among the users, and whether communication is limited per-user or only cumulatively across all users.

Many previous results in this area bound the sample and communication complexity of estimating the parameters of a distribution $P_\theta$, where $\theta \in \Theta$ (see e.g. [20]). This problem class includes discrete distribution estimation, where the guarantees are usually stated as bounds on the relative entropy or total variation distance between the estimated and true distribution (see e.g. [5]). Other problems of interest are mean estimation [37] and heavy hitter estimation [4].

There has also been significant interest in differentially private statistical estimation, and of particular relevance is the work by [3], who gave private algorithms for estimating certain functionals of a distribution, including the Shannon entropy. However, they used the central model of differential privacy, while in this paper we prove guarantees using the (stronger) local model.

## 3 Entropy Measures

The *Shannon*, *Tsallis*, and *Rényi* entropy of a discrete random variable $X$ are defined as

$$\text{(Shannon)} \qquad H(X) = -\sum_x \Pr[X = x] \log \Pr[X = x], \tag{1}$$

$$\text{(Tsallis)} \qquad T_\gamma(X) = \frac{1}{\gamma - 1}\big(1 - \sum_x \Pr[X = x]^\gamma\big), \tag{2}$$

$$\text{(Rényi)} \qquad R_\gamma(X) = \frac{1}{1 - \gamma} \log\big(\sum_x (\Pr[X = x])^\gamma\big), \tag{3}$$

where $\gamma$ in (2) and (3) is a free parameter satisfying $\gamma > 0$ and $\gamma \neq 1$. Both Tsallis and Rényi entropy are generalizations of Shannon entropy in the sense that $\lim_{\gamma \to 1} T_\gamma(X) = \lim_{\gamma \to 1} R_\gamma(X) = H(X)$.

In this paper, we describe algorithms for estimating the Shannon entropy and special cases of the Tsallis and the Rényi entropy that are widely used in many scientific fields: $T_2(X)$, also known as the *Gini entropy*, and $R_2(X)$, also known as the *collision entropy*. Substituting $\gamma = 2$ into the definitions above and using the abbreviations $G(X) \equiv T_2(X)$ and $C(X) \equiv R_2(X)$, we have:

$$\text{(Gini)} \qquad G(X) \equiv T_2(X) = 1 - \sum_x \Pr[X = x]^2,$$

$$\text{(Collision)} \qquad C(X) \equiv R_2(X) = -\log\big(\sum_x \Pr[X = x]^2\big).$$

Gini entropy is so-called because it is equivalent to the Gini diversity index, a statistic proposed by Corrado Gini in 1912 to measure income and wealth inequality [19]. Collision entropy takes its name from the observation that if $X$ and $X'$ are independent and identically distributed, then $C(X) = -\log \Pr[X = X']$.

For the problem of estimating Shannon entropy, we specialize to a high-dimensional setting, where we only observe a *pair* (or *triplet*) of the dimensions at a time. That is, $X$ is a random-vector of $d$ discrete variables, where $d$ is large, but each $X_i$ has a constant support size (e.g., they are binary), and we only observe two (or three) dimensions per sample. Without making any assumption about this joint distribution, the problem is intractable. One of the most common assumptions, which we also adopt in this work, is that the joint distribution is tree-structured. In this case, the distribution can be estimated by the celebrated [12] (and optimal [7]) Chow-Liu algorithm. While the Chow-Liu algorithm requires $\Omega(d^2)$ *pairs* observations to estimate the Shannon entropy, our sequential algorithm requires only $\mathcal{O}(d)$ *pairs* observations (see Section 5.2 for more details).

The *joint Shannon entropy* $H(X_1, \ldots, X_d)$ of a set of random variables $X_1, \ldots X_d$ is the Shannon entropy $H(X)$ of the random variable $X = (X_1, \ldots, X_d)$. We write the abbreviated term *joint entropy* when the use of Shannon entropy is obvious from context.

The *mutual information* between two random variables $X$ and $Y$ and their *conditional mutual information* given another random variable $Z$ are defined as:

$$I(X;Y) = H(X) + H(Y) - H(X,Y), \tag{4}$$
$$I(X;Y \mid Z) = H(X,Z) + H(Y,Z) - H(X,Y,Z) - H(Z). \tag{5}$$

## 4   Estimation Algorithms and Evaluation Criteria

A set of $n$ users and a central server cooperate according to the following protocol to estimate the entropy of a random variable $X$:

1. Each user $i \in [n]$ draws an independent sample $x_i$ according to the distribution of $X$.

2. For $r$ rounds:

   (a) The server sends information to a subset of the users.
   (b) Those users send (partial) information about their sample back to the server.

3. The server outputs an estimate of the Shannon entropy (Algorithms 1, 2, and 4) or the Gini or collision (Algorithm 5) entropies of $X$.

An *estimation algorithm* specifies the steps that each user and the server perform to implement the above protocol. The algorithm is *non-interactive* if the protocol consists of a single round in which all users participate. In a non-interactive algorithm the server cannot adapt its queries to users based on responses from other users, since the server communicates with all the users concurrently. An algorithm is *sequentially interactive* if each round consists of communication with a single user, who is never contacted again. Sequential interactivity enables the server to query users adaptively [22].

We evaluate estimation algorithms according to the following criteria:

- *Sample complexity*: The number of users from whom the server requests data.

- *Space complexity*: The space used by the server when executing the algorithm.

- *Communication complexity*: The maximum number of bits transmitted by any single user to the server. Note that the amount of information sent by the server to the users is not counted when determining communication complexity.

- *Privacy*: Let $x_i$ be the sample belonging to user $i$ and $o_i$ be the data observed by the server from user $i$. We say that an algorithm satisfies $\alpha$-*local differential privacy* if

$$\Pr[o_i \in O \mid x_i = x] \le e^\alpha \Pr[o_i \in O \mid x_i = x']$$

  for any user $i$, measurable set $O$, and possible sample values $x, x'$.

- *Error*: The absolute difference between the true entropy of the distribution and the estimate output by the server.

## 5   Estimating Shannon Entropy of Tree-structured Joint Distributions

In this section we assume that $X = (X_1, \dots, X_d)$ is a vector of $d$ discrete variables, and that the support size of each variable $X_i$ is constant (e.g., each variable is Boolean). We also assume that $X$ has a *tree-structured* distribution, which means that there exists a rooted tree $T$ with $d$ nodes such that for any node $i \in [d]$ we have $\Pr[X_i \mid X_{-i}] = \Pr[X_i \mid X_{\mathrm{pa}_T(i)}]$, where $X_{-i} = (X_1, \dots X_{i-1}, X_{i+1}, \dots, X_d)$ and the node $\mathrm{pa}_T(i)$ is the parent of node $i$ in tree $T$. If $i$ is the root node, then we define $\Pr[X_i \mid X_{\mathrm{pa}_T(i)}] = \Pr[X_i]$. Equivalently, a tree-structured distribution is a Markov random field with a tree as the underlying undirected graph. Essentially, the tree-structured assumption implies that the only correlations among the $X_i$'s are pairwise correlations. If $T$ is a chain or a star we say that $X$ is *chain-structured* and *star-structured*, respectively. We will treat these two special cases at the end of this section (Algorithms 2 and 4, respectively).

## 5.1 Estimating Entropy of a Marginal Distribution When the Support Size is Small

Before proceeding to describe algorithms for estimating the Shannon entropy of tree-structured distributions, we use existing results for private distribution estimation to devise a local differentially private estimator for the Shannon entropy that is sample and communication efficient when the support size of the distribution is small (as is the case for the individual marginals). The server will repeatedly invoke this algorithm as a subroutine in the sections below.

First we recall that the difference in Shannon entropy of two random variables can be upper bounded according to Theorem 17.3.3 of [13] as

$$|H(X_{\mathbf{p}}) - H(X_{\mathbf{p}'})| \leq \|\mathbf{p} - \mathbf{p}'\|_1 \log \frac{c}{\|\mathbf{p} - \mathbf{p}'\|_1}$$

where $X_{\mathbf{p}}$ and $X_{\mathbf{p}'}$ are two discrete random variables with support size $c$ and distributions $\mathbf{p}$ and $\mathbf{p}'$. Next, we apply a local differentially private learning algorithm for discrete distribution due to [4] that learns the parameters of a discrete distribution with small $L_1$ error. The following theorem combines these two results by using the fact that $x/\log(1/x) \leq 1$ whenever $0 < x \leq 1/2$.

**Theorem 5.1.** *For any discrete distribution $X$ with support size $c$ and for any $1/2 \geq \epsilon > 0$, there exists an estimator satisfying $\alpha$-local differential privacy that estimates $H(X)$ within $\epsilon$ error using $n = O(c^2 \log \frac{1}{\delta}/(\epsilon^2 \alpha^2))$ samples with probability $1 - \delta$ when $\alpha \in (0, 1)$.*

This algorithm resulting from Theorem 5.1 can be used to privately estimate the entropy $H(X_i)$, mutual information $I(X_i; X_j)$, and conditional mutual information $I(X_i; X_j \mid X_k)$ of any variables $X_i, X_j$ and $X_k$. This can be done using $O\left(\frac{\log \frac{1}{\delta}}{\alpha^2 \epsilon^2}\right)$ samples per estimate and $O(1)$ bits per sample, since each of these variables has constant support size, and both mutual information and conditional mutual information can be expressed in terms of entropies (Eqs. (4) and (5)). We call such an estimate $(\alpha, \epsilon, \delta)$-*good*.

## 5.2 Our Algorithm for Tree-structured Joint Distributions

Note that the support size of $X$ can be exponential in $d$. In the worst case, estimating the entropy of a distribution with support size $k$ within constant error requires $\tilde{\Theta}(k)$ samples [18]. However the tree-structure of $X$ can be exploited to significantly reduce the sample complexity. In their seminal paper, Chow and Liu [12] proved the identity

$$H(X) = \sum_{i=1}^{d} H(X_i) - \max_T \sum_{i=1}^{d} I(X_i; X_{\mathrm{pa}_T(i)}), \tag{6}$$

for any tree-structured random variable $X$, where the maximization is taken over all possible trees connecting the $d$ variables.

Eq. (6) suggests a communication-efficient algorithm for estimating the entropy of $X$, which is known as the *Chow-Liu algorithm*: First, estimate each marginal entropy $H(X_i)$ and each mutual information $I(X_i; X_j)$. Next, compute a maximum spanning tree on the $d$ variables, where the weight of each edge $(X_i, X_j)$ is the estimate of the mutual information $I(X_i; X_j)$. Finally, plug these estimators into Eq. (6).

The Chow-Liu algorithm requires $\Omega(d^2)$ samples, since it computes the mutual information between every pair of variables in order to compute a maximum spanning tree. However, estimating the right-hand side of Eq. (6) only requires estimating the *weight* of the maximum spanning tree, which is significantly easier than finding the tree itself. Algorithm 1 adapts a technique from [11] that estimates the weight of the maximum spanning tree of a graph in time that is sublinear in the number of edges in the graph. The basic idea is to select nodes of the graph at random and use breadth-first search to determine the size of each of their connected components if we were to drop edges that do not meet a weight threshold, short-circuiting the search when the size becomes too large. These quantities are combined to estimate the weight of the maximum spanning tree. In our case, an edge weight is a mutual information between a pair of variables, which we estimate from pair observations.

**Theorem 5.2.** *Algorithm 1 is $\alpha$-locally differentially private and has $O(1)$ communication complexity. The number of samples requested by the server is $O\left(\frac{d \log(\frac{1}{\delta})}{\alpha^2 \epsilon^3}\right)$. Let $\hat{H}$ be the entropy estimate output by the algorithm. If $X$ is tree-structured, then $|\hat{H} - H(X)| \leq \epsilon d$ with probability $1 - \delta$.*

---

**Algorithm 1** Shannon entropy estimation for tree-structured distribution

---

1: Let $M = \lceil \frac{2}{\epsilon} \rceil$ and $R = \lceil \frac{1}{\epsilon^2} \rceil$
2: **for** $m = 1, \ldots, M$ **do**
3:     **for** $r = 1, \ldots, R$ **do**
4:         Choose positive integer $Z$ randomly according to $\Pr[Z \geq z] = 1/z$.
5:         Choose $i^*$ uniformly at random from $[d]$.
6:         **if** $Z \geq \frac{2}{\epsilon}$ **then** $\gamma_{mr} \leftarrow 0$.
7:         **else**                                      ▷ Breadth-first search
8:             Initialize queue to contain $i^*$ and a set $V = \{i^*\}$.
9:             **while** queue length is non-zero and shorter than $Z$ **do**
10:                 Remove $i$ from front of queue and $V = V \cup \{i\}$.
11:                 **for** $j = [d] \setminus V$ **do**
12:                     Server computes $(\alpha, \epsilon, \delta)$-good estimate $\hat{I}_{ij}$ of $I(X_i; X_j)$.
13:                     **if** $\hat{I}_{ij} \geq \epsilon m$ **then** add $j$ to back of queue.
14:             **if** queue length is zero **then** $\gamma_{mr} \leftarrow 0$ **else** $\gamma_{mr} \leftarrow 1$.
15:     $\hat{c}_m \leftarrow \frac{d}{R} \sum_{r=1}^{R} \gamma_{mr}$.
16: $\hat{W} \leftarrow \epsilon M d - \epsilon \sum_{m=1}^{M} \hat{c}_m$
17: Server computes $(\alpha, \epsilon, \delta)$-good estimate of each entropy (for the first sum in Eq. (6)).
18: Let $\hat{S}$ be the sum of the entropy estimates.
19: Return $\hat{H} = \hat{S} - \hat{W}$.

---

## 5.3 Our Algorithm for Chain-structured Joint Distributions

Verma and Pearl [39] observed that if $X$ is chain-structured with chain $T$ then for any triplet $(X_i, X_j, X_k)$, if $X_k$ is on the unique path in $T$ between $X_i$ and $X_j$, then $I(X_i; X_j | X_k) = 0$. Thus, for any triplet $(X_i, X_j, X_k)$ in $T$, at least one of $I(X_i; X_j | X_k)$, $I(X_i; X_k | X_j)$, or $I(X_j; X_k | X_i)$ has to be zero. This observation alone is not enough to recover the chain, since the conditional mutual information $I(X_i; X_j | X_k)$ can also be zero for $X_i, X_j$ and $X_k$ when $X_k$ is not on the path between $X_i$ and $X_j$ in the chain $T$. Nevertheless, under the mild assumption that the mutual information $I(X_i, X_j)$ between every pair of variables is distinct, we can recover the chain $T$ by estimating the conditional mutual information of triplets of variables.

Our algorithm is similar to sorting algorithms such as *mergesort* [24], which require $O(d \log_2 d)$ pairwise comparisons over $d$ items. While we cannot compare pairs explicitly like in a sorting problem, for any triplet $(X_i, X_j, X_k)$, we can use their conditional mutual information estimators to locally decide which "item" is between the other two: i.e., $X_i \leftrightarrow X_j \leftrightarrow X_k$, $X_i \leftrightarrow X_k \leftrightarrow X_j$ or $X_k \leftrightarrow X_i \leftrightarrow X_j$ in the chain $T$. This suggests our Algorithm 2, which inserts the variables in a chain one by one in a sequential manner. Algorithm 2 calls Algorithm 3 as a subroutine that seeks to find the position where to insert.

**Theorem 5.3.** *Algorithm 2 is $\alpha$-locally differentially private and has $O(1)$ communication complexity. The number of samples requested by the server is $O\left(\frac{d \log \frac{d}{\delta}}{\alpha^2 \epsilon^2}\right)$. Let $\hat{H}$ be the entropy estimate output by the algorithm. If $X$ is chain-structured and $|I(X_i; X_j) - I(X_j; X_k)| \geq \epsilon$, then $|\hat{H} - H(X)| \leq \epsilon d$ with probability $1 - \delta$.*

## 5.4 Our Algorithm for Star-structured Joint Distributions

If $X$ is star-structured then recovering the star $T$ is a matter of identifying its center, which can be done by computing the mutual information between only $\tilde{\mathcal{O}}(d)$ pairs of variables. The algorithm picks a random marginal $X_i$ and takes a "Prim's step" [29], i.e., chooses the neighboring node (say $X_k$) that has the largest mutual information with $X_i$. Assuming that the mutual information $I(X_i, X_j)$ between every pair of variables is distinct, the edge between $X_i$ and $X_k$ is in the maximal spanning tree. Next, the algorithm estimates $\sum_{j \neq i} I(X_i, X_j)$ and $\sum_{j \neq k} I(X_k, X_j)$ to decide whether $X_i$ or $X_k$ is the center node of the star. Algorithm 4 presents the procedure, and Theorem 5.4 gives its sample complexity.

**Theorem 5.4.** *Algorithm 4 is $\alpha$-locally differentially private and has $O(1)$ communication complexity. The number of samples requested by the server is $O\left(\frac{d \log \frac{d}{\delta}}{\alpha^2 \epsilon^2}\right)$. Let $\hat{H}$ be the entropy estimate output by the algorithm. If $X$ is star-structured and $|I(X_i; X_j) - I(X_j; X_k)| \geq \epsilon$, then $|\hat{H} - H(X)| \leq \epsilon d$ with probability $1 - \delta$.*

---

**Algorithm 2** Shannon entropy estimation for chain-structured distribution

---

1: $S = [d]$, $C = \emptyset$, pick an arbitrary $i, j, k \in S$ and set $S = S \setminus \{i, j, k\}$.
2: Server computes $(\alpha, \epsilon, \delta)$-good estimates $\hat{I}(X_i; X_j \mid X_k)$, $\hat{I}(X_i; X_k \mid X_j)$ and $\hat{I}(X_k; X_j \mid X_i)$.
3: **if** $\hat{I}(X_i; X_j \mid X_k) > \epsilon$ **then** $\boldsymbol{x}_1 = (i, k, j)$
4: **else if** $\hat{I}(X_i; X_k \mid X_j) > \epsilon$ **then** $\boldsymbol{x}_1 = (i, j, k)$
5: **else if** $\hat{I}(X_k; X_j \mid X_i) > \epsilon$ **then** $\boldsymbol{x}_1 = (j, i, k)$
6: **for** $i \in (1, \ldots, d-3)$ **do**
7:     Pick item $j$ from $S$ and set $S = S \setminus \{j\}$, $r = x_{i,1}$ and $p = x_{i,i+2}$
8:     Server computes $(\alpha, \epsilon, \delta)$-good estimates $\hat{I}(X_j; X_p \mid X_o)$, $\hat{I}(X_r; X_j \mid X_p)$.
9:     **if** $\hat{I}(X_j; X_p \mid X_r) > \epsilon$ **then** $\boldsymbol{x}_{i+1} = (j, \boldsymbol{x}_i)$     ▷ Attach $X_j$ to the head of the chain
10:     **else if** $\hat{I}(X_o; X_j \mid X_p) > \epsilon$ **then** $\boldsymbol{x}_{i+1} = (\boldsymbol{x}_i, j)$     ▷ Attach $X_j$ to the tail of the chain
11:     **else**     ▷ Insert $X_j$ into the chain defined by $\boldsymbol{x}_i$
12:         $\ell = \text{TERNARYSEARCH}(\boldsymbol{x}_i, 1, i+2, j)$     ▷ Defined in Algorithm 3
13:         $\boldsymbol{x}_{i+1} = (\boldsymbol{x}_i[1, \ldots, \ell], j, \boldsymbol{x}_i[\ell+1, \ldots, i+2])$
14: Create chain $T$ according to the order defined by $\boldsymbol{x}_{d-2}$.
15: Server computes $(\alpha, \epsilon, \delta)$-good estimate of each term in Eq. (6) using $T$ and returns $\hat{H}$.

---

**Algorithm 3** TERNARYSEARCH$(\boldsymbol{x}, \ell_l, \ell_h, j)$

---

1: **if** $\ell_l = \ell_h - 1$ **then return** $\ell_l$
2: Pick the median element $k = \lceil (\ell_h + \ell_l) \rceil$, and set $i = x_{\ell_l}$ and $o = x_{\ell_h}$
3: Server computes $(\alpha, \epsilon, \delta)$-good estimate $\hat{I}(X_i; X_k \mid X_j)$.
4: **if** $\hat{I}(X_i; X_k \mid X_j) > \epsilon$ **then return** TERNARYSEARCH$(\boldsymbol{x}, i, k, j)$
5: **else return** TERNARYSEARCH$(\boldsymbol{x}, k, o, j)$

---

**Algorithm 4** Shannon entropy estimation for star-structured distribution

---

1: Pick $i \in [d]$ uniformly at random.
2: Server computes $(\alpha, \epsilon, \delta)$-good estimate $\hat{I}(X_i, X_j)$ for all $j \in [d] \setminus \{i\}$.
3: Find $k = \arg\max_{j \in [d] \setminus \{i\}} \hat{I}(X_i, X_j)$
4: Server computes $(\alpha, \epsilon, \delta)$-good estimate $\hat{I}(X_k, X_j)$.
5: **if** $\sum_j \hat{I}(X_i, X_j) > \sum_j \hat{I}(X_k, X_j)$ **then** let $T$ be a star with $X_i$ as center
6: **else** let $T$ be a star with $X_k$ as the center.
7: Server computes $(\alpha, \epsilon, \delta)$-good estimate of each term in Eq. (6) using $T$ and returns $\hat{H}$.

---

## 5.5 Lower Bounds

We prove sample complexity lower bounds for estimating the Shannon entropy of a tree-structured joint distribution from pair observations. Our first lower bound focuses on the non-interactive case, when the algorithm must select all the pairs in advance. The second claim is more general, and holds for all sequentially interactive algorithms.

**Theorem 5.5.** *For any non-interactive algorithm that uses $o(d^2)$ pair observations to estimate Shannon entropy, there exists a tree-structured distribution over $\{0, 1\}^d$ such that the error of the algorithm is $\Omega(d)$ with constant probability.*

**Theorem 5.6.** *For any $\epsilon > 0$ and for any sequentially interactive algorithm that uses $o(d/\epsilon)$ pair observations to estimate Shannon entropy, there exists a tree-structured distribution on $\{0, 1\}^d$ such that the error of the algorithm is $\Omega(\epsilon \cdot d)$ with constant probability.*

The lower bound given in Theorem 5.5 is based on Turán's theorem [8], which we use to show that for any algorithm with sub-quadratic sample complexity and for any constant $C \in (0,1)$, there is a graph with $d$ nodes containing a $C \cdot d$-clique – when $d$ is large enough – such that the algorithm does not observe any edge of that clique. This implies that the additive error of the algorithm is linear in $d$. The lower bound for sequentially interactive algorithms in Theorem 5.6 is based on a information theoretical approach. Interestingly, our construction of problem instances for which we applied Le Cam's theorem is fairly simple, since it contains $d$ independent random variables in every case. Nevertheless, this lower bound shows that Algorithm 1 is optimal in $d$.

### 5.6 Comparison to Prior Work

To the best of our knowledge, the Chow-Liu algorithm is the only published method for estimating the entropy of a distribution that takes advantage of its tree structure. Since the algorithm is non-interactive, the lower bound in Theorem 5.5 shows that our algorithms have provably better sample complexity when the number of variables $d$ is large (note that the dependence on $d$ in each of Theorems 5.2, 5.3 and 5.4 is sub-quadratic). The Chow-Liu algorithm can also be used to estimate the distribution itself, not just its entropy, and it has recently been shown [7, 15] that the algorithm has optimal sample complexity when given full observations (i.e., samples of the entire vector $(X_1, \ldots, X_d)$ and not just pairs or triplets of the variables). Thus the Chow-Liu algorithm is optimal for estimating a tree-structured distribution, but suboptimal for estimating the *entropy* of a tree-structured distribution. The root cause of this difference appears to be the fact that it is significantly easier to estimate the weight of the maximum spanning tree than finding the tree itself.

## 6 Estimating Gini and Collision Entropies

In this section, we describe a non-iterative protocol (Algorithm 5) that estimates both the Gini and collision entropies of a discrete random variable $X$ while observing only $b$ bits per sample from its distribution, and does not require any extra assumption on the distribution of $X$ (such as assuming it is tree-structured). First, the server partitions all users into pairs (assume for simplicity that the number of users is even). The server then distributes a $b$-bit hash function to each user, along with a distinct salt to each user pair. Each user then hashes their sample along with their salt, and returns the hash value to the server. The server computes entropy estimates based on the number of observed hash collisions across all pairs. In Algorithm 5, we let $\langle x, y \rangle$ denote a binary string that encodes $x$, followed by a delimiter, and by $y$.

---

**Algorithm 5** Gini and collision entropy estimation

---

1: Each user $i \in [n]$ draws a sample $x_i$ independently from the distribution of $X$.
2: Server partitions the $n$ users into $\frac{n}{2}$ disjoint pairs.
3: Let $q_i \in \left[\frac{n}{2}\right]$ be the index of the pair containing user $i$.
4: Server transmits $q_i$ and hash function $h : \{0,1\}^* \mapsto \{0,1\}^b$ to each user $i$.
5: Each user $i$ generates a $b$-bit hash value $h_i = h(\langle q_i, x_i \rangle)$ for their sample $x_i$.
6: Each user $i$ lets $\hat{h}_i = h_i$ with probability $\lambda = \frac{e^\alpha}{2^b + e^\alpha}$ and otherwise draws $\hat{h}_i$ uniformly from $[2^b]$.

7: Server receives $\hat{h}_i$ from each user $i$.
8: If pair $q$ contains users $i$ and $j$ then let $c_q = \mathbf{1}[\hat{h}_i = \hat{h}_j]$ indicate whether a hash collision was observed for pair $q$.
9: Server computes $\bar{c} = \left( \frac{2}{(1-\lambda)n} \sum_q c_q \right) - \frac{\lambda}{2^b}$.
10: Server outputs $\hat{G} = \frac{2^b}{2^b - 1} \bar{c} - \frac{1}{2^b - 1}$ and $\hat{C} = -\log\left(1 - \hat{G}\right)$.

---

Algorithm 5 is based on the observation that if $X$ and $X'$ are independent and identically distributed then the Gini entropy is equal to $1 - \Pr[X = X']$ and the collision entropy is equal to $-\log \Pr[X = X']$. If the server observed each sample directly then it could estimate $\Pr[X = X']$ using the collision frequency, i.e., the fraction of sample pairs $(x_i, x_j)$ such that $x_i = x_j$. However, the server only observes a $b$-bit hash of each sample. Among sample pairs in which there is a true collision, all of them also produce a hash collision. Among samples pairs in which there is not a true collision, about a $\frac{1}{2^b}$ fraction of them produce a hash collision. Therefore the true collision frequency can be

estimated using an appropriately bias-corrected hash collision frequency, and the server uses this estimate to approximate the Gini and collision entropies.

The analysis of Algorithm 5 is given in Theorem 6.1 below. As is customary, for the analysis we assume that the hash function $h$ is constructed by assigning each element of its domain to a uniform random element of its range [6]. See the Appendix G for the proof of the theorem.

**Theorem 6.1.** *Algorithm 5 is $\alpha$-locally differentially private, has $\tilde{O}(b)$ communication complexity and $\tilde{O}(b)$ space complexity. Let $\hat{G}$ and $\hat{C}$ be the outputs of the algorithm. Let $\alpha, \epsilon, \delta \in (0, 1)$. If $n = \Omega\left(\frac{b^2 \max\{1-G(X), 2^{-b}\} \log \frac{1}{\delta}}{\alpha^2 \epsilon^2}\right)$, then $|\hat{G} - G(X)| \leq \epsilon$ with probability at least $1 - \delta$. Also, if $X$ has support size $k$ and $n = \Omega\left(\frac{b^2 k^2 \log \frac{1}{\delta}}{\alpha^2 \epsilon^2 \min\{k, 2^b\}}\right)$, then $|\hat{C} - C(X)| \leq \epsilon$ with probability at least $1 - \delta$.*

### 6.1 Comparison to Prior Work

Recall that Gini entropy is proportional to the second frequency moment. Local differentially private algorithms for estimating frequency moments were recently studied in [10]. Letting $b = 1$ in Algorithm 5 yields a sample complexity of $\tilde{O}(1/\alpha^2 \epsilon^2)$, which is independent of the distribution's support size, unlike the sample complexity of the non-interactive algorithm for estimating the second frequency moment from [10]. Also our algorithm only uses 1 bit per sample and $\tilde{O}(1)$ space, while the previous algorithm uses $\Omega(k)$ bits per sample and $\Omega(k)$ space, where $k$ is the support size of the distribution. The authors in [10] asked whether there is a non-interactive algorithm for privately estimating frequency moments with a sample complexity that is independent of the distribution's support size. Here we affirmatively answer this open question for the second frequency moment.

The best known algorithm for estimating collision entropy using $\tilde{O}(1)$ space is due to [36]. The sample complexity of their algorithm is $\tilde{O}\left(k/\epsilon^2\right)$ and its communication complexity is $O(\log k)$ bits per user. Letting $b = \log k$ and $\alpha = O(1)$ in Algorithm 5 recovers these results (up to logarithmic factors), and using smaller values for $b$ or $\alpha$ generalizes the previous algorithm to the private and communication-efficient setting. Also, it was shown in [14] that (conditioned on a plausible conjecture) any algorithm that estimates collision entropy to within $O(1)$ error using $O(1)$ space requires $\Omega(k)$ samples. Therefore our algorithm is likely to be Pareto optimal with respect to the sample complexity-space complexity trade-off.

## 7 Experiments

In this section we present two sets of experiments to support our theoretical findings. First, we demonstrate that Algorithm 1 is indeed able to estimate the Shannon entropy of tree-structured distributions with linear sample complexity in $d$. Thus it has a superior sample complexity comparing to the state-of-the-art non-interactive method [12, 7], which has a quadratic sample complexity in $d$. The sample complexity is defined here in terms of number of observations from pairs of marginals. In the second set of experiments, we use our Algorithm 5 to estimate the collision entropy of discrete distributions, and compare its performance to that of the best-known communication efficient, non-private algorithm for this task (which we refer to as Skorski's algorithm [36]).

**Estimating Shannon entropy:** To estimate the Shannon entropy of a tree-structured distribution as given by eq. (6), the marginal entropies and the mutual information between certain or all pairs of marginals have to be estimated. The Chow-Liu algorithm estimates the mutual information between all pairs of marginals which results in quadratic sample complexity, whereas Algorithm 1 estimates the mutual information only for a linear fraction of pairs. Both algorithms estimate the marginal entropy values by sampling the marginals independently from the mutual information estimations, and they both use the same $\epsilon$ additive error and privacy budget $\alpha$ to estimate the mutual information between pairs of marginals. Thus it is fair to compare their performance in terms of number of pairs for which they estimate the mutual information. For the Chow-Liu algorithm this is always $d^2$, whereas our Algorithm 1 is randomized, thus we evaluate it over 100 repetitions and report the average.

We ran this experiment on random tree-structured joint distributions over $\{0, 1\}^d$. To create a random tree-structured distribution, we first sample the skeleton of the distribution by taking the maximum spanning tree of a complete graph with $d$ nodes and edge weights distributed according to a standard normal. Next, we sample the parameters for each marginal uniformly at random from $[0, 1]$, and

then we achieve the tree-structure by inducing dependence between pairs of variables while preserving the marginals. Specifically, for two marginals $X_i$ and $X_j$ with sampled parameters $p_i$ and $p_j$, we set: $P(X_i = 0, X_j = 0) = (1 - p_i) \cdot (1 - p_j) + r_{ij}$, $P(X_i = 1, X_j = 1) = p_i \cdot p_j + r_{ij}$, $P(X_i = 0, X_j = 1) = (1 - p_i) \cdot p_j - r_{ij}$, and $P(X_i = 1, X_j = 0) = p_i \cdot (1 - p_j) - r_{ij}$, where $r_{ij}$ is sampled uniformly at random from a range of values such that each probability stays positive. The results are displayed in Figure 1a. It is clear that the sample size (i.e., number of mutual information estimate) is close to linear for our algorithm, whereas the Chow-Liu algorithm requires a larger sample size.

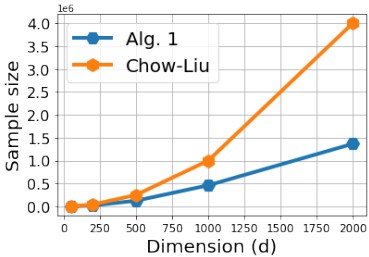

(a) Sample complexity of estimating Shannon entropy of tree-structured distributions (eq. (6)) for the Chow-Liu algorithm [12] and for our Algorithm 1.

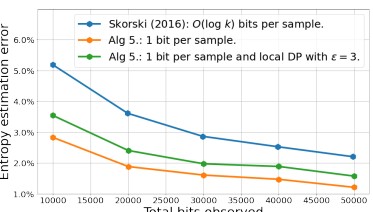

(b) Absolute error in estimating the collision entropy of an exponential distribution with domain size $k = 1000$ for the Skorski's algorithm [36] and for our Algorithm 5.

**Estimating collision entropy:** In this set of experiments, we drew samples from a discrete exponential distribution $p_i \propto e^{-i}$ with support size $k = 1000$ and estimated the collision entropy using algorithm [36] (the previously best-known communication-efficient algorithm for this task) and our Algorithm 5 (with and without local differential privacy). Algorithm [36] requires $O(\log k)$ bits per sample and is not private, while our algorithm only requires 1 bit per sample and is differentially private. The results are displayed in Figure 1b. The previous algorithm has $5\%$ estimation error after observing 10000 bits, while our algorithm has less than $3.5\%$ estimation error. Thus our algorithm has lower error for the same communication cost while also being local differentially private.

## 8 Conclusion and future work

Estimating entropy is of importance in many practical applications. In this paper, we studied three widely used entropy measures: Shannon, Gini and collision entropy. We described estimation algorithms for each entropy that require minimal communication and satisfy local differential privacy. We also validated our theoretical results with simulations.

Our sequentially interactive algorithm for estimating Shannon entropy of high-dimensional tree-structured distributions observes only two of these dimensions per sample and has a sample complexity $O(d/\epsilon^3)$. Our approach relies on the celebrated Chow-Liu approximation [12], providing a substantial improvement on the $\Omega(d^2)$ sample complexity of the original non-interactive Chow-Liu algorithm. We also identified two special cases (viz., when the underlying graphical model of the joint distribution is either a chain or star graph) and provided algorithms with a sample complexity of $\tilde{O}(d \log d/\epsilon^2)$. Our algorithm for Gini and collision entropy estimation also improved on the state-of-the-art, either by improving the sample complexity and communication complexity of previous work, or by generalizing the best known algorithm to the private and communication-efficient setting.

A natural extension of our work on Shannon entropy estimation is to consider higher-order correlations in the Chow-Liu decomposition [25]. In contrast to the second-order case which reduces to a maximum spanning tree problem, discovering the underlying structure of the joint distribution is already computationally challenging. However, efficiently estimating the entropy of the resulting distribution might still be possible.

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
