## A Proof of Theorem 5.1

*Proof.* [4, 5] showed that any discrete distribution can be learnt in total variation distance based on $O(c^2 \log 1/\delta/(\epsilon^2\alpha^2))$ when $\alpha < 1$. This result can be plug-in into Theorem 17.3.3 of [13]. $\quad\square$

## B Proof of Theorem 5.2

The result follows from simply combining the algorithm from [11] for estimating the weight of the maximum spanning tree in sublinear time, which assumes that each edge weight can be computed in $O(1)$ time, with Theorem 5.1, which gives the number of samples needed to privately estimate an edge weight (i.e., the mutual information between two variables). The main technical complication we must overcome is that the result from [11] assumes that the edge weights are integers with a bounded ratio, while we instead discretize the edge weights with resolution $\epsilon$. To complete the proof, we need the Lemma B.1 below, which concerns a graph $G$ whose edge weights are multiples of $\epsilon$. The lemma relates the weight of the maximum spanning tree of $G$ to the number of connected components in various subgraphs of $G$. This lemma replaces Claim 5 from [11].

Let $G$ be a connected graph with $n$ vertices and edge weights belonging to the set

$$\{\epsilon, 2\epsilon, \ldots, w\epsilon\},$$

where $\epsilon > 0$ and $w$ is a positive integer.

For each $i \in \{1, \ldots, w\}$ let $G_i$ be the subgraph of $G$ containing the same vertices as $G$ and only those edges of $G$ whose weight is at least $i\epsilon$. Thus $G_1 = G$. Let $c_i$ be the number of connected components in $G_i$. Let $M$ be the weight of a maximum spanning tree of $G$.

**Lemma B.1.** $M = \epsilon wn - \epsilon \sum_{i=1}^{w} c_i$.

*Proof.* For each $i \in \{1, \ldots, w\}$ let $\gamma_i$ be the number of edges with weight $i\epsilon$ in a maximum spanning tree of $G$. We have

$$\sum_{i < \ell} \gamma_i = c_\ell - 1$$

for any $\ell \in \{1, \ldots, w\}$, where the empty sum is defined to be zero. This equality can be established by considering Kruskal's greedy algorithm for constructing a maximum spanning tree, which adds edges in decreasing order of weight as long as they do not induce a cycle. Since $G_\ell$ has $c_\ell$ connected components, and all the edges in $G_\ell$ are heavier than all the edges not in $G_\ell$, the greedy algorithm must first connect the vertices within each component of $G_\ell$ and then use exactly $c_\ell - 1$ edges not in $G_\ell$ to connect the components to each other. Thus we have

$$M = \sum_{i=1}^{w} i\epsilon\gamma_i$$

$$= \epsilon\left(\sum_{i\geq 1}\gamma_i + \sum_{i\geq 2}\gamma_i + \cdots + \sum_{i\geq w}\gamma_i\right)$$

$$= \epsilon\left(n-1-\sum_{i<1}\gamma_i + n-1-\sum_{i<2}\gamma_i + \cdots + n-1-\sum_{i<w}\gamma_i\right)$$

$$= \epsilon\left(w(n-1) - \sum_{i=1}^{w}(c_i - 1)\right)$$

$$= \epsilon wn - \epsilon\sum_{i=1}^{w}c_i \qquad\qquad\square$$

## C Proof of Theorem 5.3

*Proof.* We start by recalling a lemma that applies to tree-structured distributions.

**Lemma C.1.** *[39] Let* $\mathbf{X} = (X_1, \ldots, X_d)$ *be tree decomposable with tree* $T$. *Then for any triplets* $i$, $j$ *and* $k$, *if* $k$ *is on the unique path in* $T$ *between* $i$ *and* $j$, *then*

$$I(X_i; X_j | X_k) = 0 \ .$$

Next we show that the output Algorithm 2 is correct with high probability. We make use of the Conditional Mutual Information Tester of [7]. This testing algorithm consists of estimating the CMI using the plug-in estimator and then applying a $\epsilon$ threshold on the estimate, i.e., if the estimate is smaller than $\epsilon$ then accept, otherwise reject. The sample complexity of Conditional Mutual Information Tester is $O\left(\frac{|\Sigma|^3}{\epsilon} \log \frac{d|\Sigma|}{\delta} \log \frac{|\Sigma| \log d/\delta}{\epsilon}\right)$ according to Theorem 1.3 of [7]. Thus if we apply this tester with adjusted confidence parameter, i.e., $\delta/d \log_3 d$, then the union bound implies that the output of all tests is correct with probability at least $1 - \delta$.

Next, note that for any triplet $X_i, X_j, X_k$ such that $X_i$ is between $X_j$ and $X_k$ in the chain, it holds that

$$I(X_j; X_i) - I(X_j; X_k) = \underbrace{I(X_j; X_i | X_k)}_{> \epsilon} - \underbrace{I(X_j; X_k | X_i)}_{=0 \text{ due to Lemma C.1}}$$

as the edges are different with a margin of $\epsilon$. The same argument implies that $I(X_k; X_i | X_j) > \epsilon$. Thus, Algorithm 2 divides the nodes correctly in Line 4-10 which along with the testers' correctness with high probability implies the correctness of the algorithm.

Using the result from [7] that the sample complexity of Conditional Mutual Information Tester is $O\left(\frac{|\Sigma|^3}{\epsilon} \log \frac{d|\Sigma|}{\delta} \log \frac{|\Sigma| \log d/\delta}{\epsilon}\right)$, we only need to upper bound the number of tests for deciding whether $I(X_i; X_j \mid X_k) > \epsilon$ that is carried out by Algorithm 2. It is indeed $3 \sum_{i=3}^{d} \log_3 i \in O(d \log_3 d)$ which concludes the proof. $\quad\square$

# D   Proof of Theorem 5.4

*Proof.* First note that the Prim step in Line 3 of Algorithm 4 indeed finds an edge that is in the maximum spanning tree due to the assumption $|I(X_i; X_j) - I(X_j; X_k)| \geq \epsilon$. Say this edge is between $X_i$ and $X_k$, what remains to be decided is whether $X_i$ or $X_k$ is the center of the graph. This can be done by comparing $\sum_j \hat{I}(X_i, X_j)$ and $\sum_j \hat{I}(X_k, X_j)$.

So Algorithm 4 estimates $2d$ mutual information, which requires $4d$ marginal and $2d$ pairwise marginal entropy estimation. Recall from Theorem 5.1 that entropy can estimated $\alpha$-locally differential private with $\epsilon$ error using $O(c^2 \log \frac{1}{\delta}/(\epsilon^2 \alpha^2))$ samples. Thus by the union bound the algorithm is correct if the confidence parameter is set to $\delta/6d$, and accordingly has sample complexity $O(\frac{6dc^2}{\epsilon^2\alpha^2} \log \frac{6d}{\delta})$. $\quad\square$

# E   Proof of Theorem 5.5

*Proof.* Assume a deterministic algorithm $\mathcal{A}$ that takes sub-quadratic samples from $\mathbf{X}$ and estimates $H(\mathbf{X})$. In addition, assume that its sample complexity is $o(d^2)$. Thus for any constant $C > 0$, there exists $d_0$ such that for any $d > d_0$, the sample complexity of the algorithm is $< Cd^2$. This implies that, if $d$ is large enough, the algorithm needs $\leq d^{2-\kappa}$. In addition, we can pick $d$ so as $d^{-\kappa} < \kappa$ which implies $d^2 - d^{2-\kappa} > (1 - \kappa)d^2$. Thus any deterministic algorithm which takes sub-quadratic sample size never observes $(1 - \kappa)d^2$ edges for $\forall \kappa$ when $d$ is large enough.

Let us recall that Turán's theorem

**Theorem E.1** ([8]). *Let* $G(V, E)$ *be a graph with* $d$ *vertices (i.e,* $|V| = d$*) that does not contain a* $(\ell + 1)$-*clique as a subgraph. Then* $G$ *has at most* $\frac{(\ell-1)d^2}{2\ell}$ *edges.*

In addition, recall that a Turán's graph $G_T(d, \ell)$ [8] is defined as the unique graph with $d$ nodes that does not contain a $(\ell + 1)$-clique and has the maximum possible number of edges which is $\left\lfloor \frac{(\ell-1)d^2}{2\ell} \right\rfloor$. Let $t(d, \ell)$ denote the number of edges in $G_T(d, \ell)$.

Thus for any $d > 0$ there exists a graph $G = (V, E)$ such that $|V| = d$ and $|E| > t(d, k)$ which contains a $(k + 1)$-clique. This implies that if algorithm $\mathcal{A}$ does not observes at least $t(d, k)$ edges of $G$, then $G$ contains a $k + 1$-clique whose edges are never observed by algorithm $\mathcal{A}$. Now we apply Turan's result with $\ell = d/2$ which implies that

$$t(d, d/2) = \left\lfloor (1/2 - 1/d)d^2 \right\rfloor$$

Thus for any $\kappa \in (0, 1/2)$ and any algorithm with sample complexity $o(d^2)$, if $d$ is large enough, then there will be a $\Theta(d/2)$-clique for which the algorithm does not observe any edge within this clique.

Finally it is easy to construct two $d/2$-dimensional problem instances, denoted it $S$ and $S'$, with joint entropy that differs by $\Omega(d)$: take $d/2$ Bernoulli with parameter $1/2$ and take the copy of the same Bernoulli $d/2$ times. The entropy for these two joint distributions is $d/2$ and $1$, respectively. This also implies that for any deterministic algorithm we can construct two problem instances which contains $S$ and $S'$ so as they are independent from the rest of the marginals and the algorithm does not observe any sample from them, and hence it cannot achieve $o(d)$ additive error. $\qquad\square$

## F   Proof of Theorem 5.6

*Proof.* Let $\hat{\theta}_n = \hat{\theta}(x_1, \ldots, x_n)$ such that $\hat{\theta}_n : (\Sigma^d)^n \mapsto \mathbb{R}$ be an estimator using $n$ samples.

**Theorem F.1.** *[Le Cam's theorem] Let $\mathcal{P}$ be a set of distributions. Then, for any pair of distributions $P_0, P_1 \in \mathcal{P}$, we have*

$$\inf_{\hat{\theta}} \max_{P \in \mathcal{P}} \mathbb{E}_P \left[ d(\hat{\theta}_n(P), \theta(P)) \right] \geq \frac{d(\theta(P_0), \theta(P_1))}{8} e^{-n d_{KL}(P_0, P_1)},$$

*where $\theta(P)$ is a parameter taking values in a metric space with metric $d$, and $\hat{\theta}_n$ is the estimator of $\theta$ based on $n$ samples.*

Let us consider two Bernoulli distributions $P_0$ and $P_1$ with parameters $p_0 = 1/2$ and $p_1 = 1/2 - \epsilon$, where $\epsilon \in (0, 1/2)$. The entropy of random variables $X_0$ and $X_1$ distributed according to $P_0$ and $P_1$ are $H(X_0) = 1$ and

$$H(X_1) = - \left( \frac{1}{2} - \epsilon \right) \log_2 \left( \frac{1}{2} - \epsilon \right) - \left( \frac{1}{2} + \epsilon \right) \log_2 \left( \frac{1}{2} + \epsilon \right).$$

Thus,

$$
\begin{aligned}
|H(X_0) - H(X_1)| = H(X_0) - H(X_1) &= \left( \epsilon + \frac{1}{2} \right) \log_2(1 + 2\epsilon) - \left( \epsilon - \frac{1}{2} \right) \log_2(1 - 2\epsilon) \\
&\geq \left( \epsilon + \frac{1}{2} \right) \frac{2\epsilon}{1 + 2\epsilon} - \left( \frac{1}{2} - \epsilon \right) 2\epsilon \\
&\geq 2\epsilon^2
\end{aligned}
$$

where we used that $\log(1 - 2\epsilon) \leq -\epsilon$ for $0 < \epsilon < 1$ and $\frac{\epsilon}{1+\epsilon} \leq \log(1 + \epsilon)$ for $\epsilon > -1$. The KL divergence can be upper bounded as

$$d_{\mathrm{KL}}(P_0, P_1) = -\frac{1}{2} \log_2(1 - 4\epsilon^2) \leq 2\epsilon^2.$$

We can now apply the Le Cam's theorem for the set of Bernoulli distributions with metric $d$ being the $\ell_1$-norm as

$$\inf_{\hat{\theta}} \max_{P \in \mathcal{P}} \mathbb{E}_P \left[ |\hat{\theta}_n(P) - H(P)| \right] \geq \frac{d(\theta(P_0), \theta(P_1))}{8} e^{-n d_{\mathrm{KL}}(P_0, P_1)} \geq \frac{\epsilon^2}{4} e^{-2n\epsilon^2}$$

Using this result with $\epsilon' = \sqrt{\epsilon}$, the following sample complexity can be obtained for estimating Shannon entropy.

**Corollary F.2.** *For any $\hat{\theta}_n$ such that $n \in o(1/\epsilon)$, there exists a Bernoulli distribution $P$ for which*

$$\mathbb{E}_P \left[ |\hat{\theta}_n(P) - H(P)| \right] \geq C \cdot \epsilon,$$

*with $C > 0$ that does not depend on $\epsilon$.*

First, note that there is some bound on error $r(\delta)$, either lower or upper, that holds with probability $1 - \delta$. This translate into the bound $r(\delta) + \delta$ on the expected error in a straightforward manner. Thus the lower bound presented in Corollary F.2 also implies that there is no high probability estimator for entropy with $o(1/\epsilon)$ sample complexity for discrete distributions. This can be used to lower bound of the entropy estimator for joint distribution as follows.

Let $\mathcal{B} = \{\mathbf{b} = (b_1, \ldots, b_d) : b_j \in \{0, 1\}\}$ be the vertices of the $d$ dimensional hypercube, and define a set of $d$-dimensional distribution $\mathcal{P}_{\mathbf{b}}$ indexed by the element of $\mathcal{B}$. Each $P_{\mathbf{b}} \in \mathcal{P}$ contains $X_0 \sim \text{Bern}(1/2)$ if $b_i = 0$ and $X_1 \sim \text{Bern}(1/2 - \epsilon)$ if $b_i = 1$, i.e.

$$P_{\mathbf{b}} = X_{b_1} \oplus \cdots \oplus X_{b_d}$$

and

$$\mathcal{P} = \left\{ P_{\mathbf{b}} : \mathbf{b} \in \{0, 1\}^d \right\} .$$

It is clear that $\mathcal{P}$ is a subset of the tree-structured distributions and each distribution contains $d$ independent Bernoulli random variables, thus

$$H(P_{\mathbf{b}}) = \sum_{i=1}^{d} H(X_{b_i})$$

Therefore any estimator that achieves at most $\epsilon \cdot d$ additive error for $H(P_{\mathbf{b}})$ has to estimate each individual Bernoulli distribution with at most $\epsilon$ error. The sample complexity of any estimator of $H(P_{\mathbf{b}})$ with an additive error $O(\epsilon d)$ is $\Omega(d/\epsilon)$. □

# G    Proof of Theorem 6.1

*Proof.* Algorithm 5 clearly has $\tilde{O}(b)$ communication complexity and $\tilde{O}(b)$ space complexity, since it only has to maintain a counter of collisions between $b$-bit hashes. Each user replaces their hash with a random hash with probability $\lambda$, and therefore the algorithm is $\alpha$-local differentially private, since $\log\left(\frac{\lambda}{(1-\lambda)/2^b}\right) = \log\left(\frac{\lambda 2^b}{1-\lambda}\right) = \alpha$ where we used $\lambda = \frac{e^\alpha}{e^\alpha + 2^b}$. Before proving the sample complexity we will first prove the following result.

**Lemma G.1.** *Let $\hat{G}$ be output by Algorithm 5. Let $\epsilon, \delta \in (0, 1)$. If*

$$n \geq \frac{6b^2 \log \frac{2}{\delta}}{\alpha^2 \epsilon^2 \left(\left(1 - \frac{1}{2^b}\right) G(X) + \frac{1}{2^b}\right)}$$

*then $|\hat{G} - G(X)| \leq \epsilon\left(G(X) + \frac{1}{2^b - 1}\right)$ with probability at least $1 - \delta$.*

*Proof.* Recall that if $X$ and $X'$ are independent and identically distributed then

$$G(X) = 1 - \Pr[X = X'].$$

We will calculate the expected value of each $c_q$. Suppose pair $q$ contains samples $x_i$ and $x_j$. If $x_i = x_j$ then $c_q = 1$ with probability $1 - \lambda + \frac{\lambda}{2^b}$, and otherwise $c_q = 1$ with probability $\frac{1}{2^b}$. Thus

$$\begin{aligned}
\mathrm{E}[c_q] &= \Pr[c_q = 1 \mid x_i = x_j] \Pr[x_i = x_j] + \Pr[c_q = 1 \mid x_i \neq x_j] \Pr[x_i \neq x_j] \\
&= \left(1 - \lambda + \frac{\lambda}{2^b}\right) \Pr[x_i = x_j] + \frac{1}{2^b} \Pr[x_i \neq x_j] \\
&= \left(1 - \lambda + \frac{\lambda}{2^b}\right) \Pr[x_i = x_j] + \frac{1}{2^b}(1 - \Pr[x_i = x_j]) \\
&= \left(1 - \lambda + \frac{\lambda}{2^b} - \frac{1}{2^b}\right) \Pr[x_i = x_j] + \frac{1}{2^b} \\
&= \left(1 - \lambda + \frac{\lambda}{2^b} - \frac{1}{2^b}\right) (1 - G(X)) + \frac{1}{2^b}
\end{aligned}$$

where the last line follows because $x_i$ and $x_j$ independent samples from the distribution of $X$.

Recall that by the Chernoff bound if $z_1, \ldots, z_m$ are independent random variables such that $z_i \in \{0, 1\}$ then for all $\epsilon \in (0, 1)$ the average $\bar{z} = (z_1 + \cdots z_m)/m$ satisfies

$$\Pr\left[\bar{z} \geq (1 + \epsilon)\operatorname{E}[\bar{z}]\right] \leq \exp\left(-\frac{\epsilon^2 m}{3}\operatorname{E}[\bar{z}]\right), \text{ and}$$

$$\Pr\left[\bar{z} \leq (1 - \epsilon)\operatorname{E}[\bar{z}]\right] \leq \exp\left(-\frac{\epsilon^2 m}{3}\operatorname{E}[\bar{z}]\right).$$

The $c_q$'s are independent random variables because each $c_q$ is defined using a distinct pair of samples and distinct pair index. Also, each $c_q \in \{0, 1\}$. Note that we proved above that the average of the $c_q$'s is $\frac{2}{n}\sum_q \operatorname{E}[c_q] = \left(1 - \lambda + \frac{\lambda}{2^b} - \frac{1}{2^b}\right)(1 - G(X)) + \frac{1}{2^b}$. Therefore

$$
\begin{aligned}
\Pr\left[\hat{G} \geq G(X) + \epsilon\left(G(X) + \frac{1}{2^b - 1}\right)\right] &= \Pr\left[\frac{2^b}{2^b - 1}\bar{c} - \frac{1}{2^b - 1} \geq G(X) + \epsilon\left(G(X) + \frac{1}{2^b - 1}\right)\right] \\
&= \Pr\left[\bar{c} \geq (1 + \epsilon)\left(\left(1 - \frac{1}{2^b}\right)G(X) + \frac{1}{2^b}\right)\right] \\
&= \Pr\left[\bar{c} \geq (1 + \epsilon)\operatorname{E}[\bar{c}]\right] \\
&\leq \exp\left(-\frac{\epsilon^2 n}{6}\operatorname{E}[\bar{c}]\right) \\
&= \exp\left(-\frac{\alpha^2 \epsilon^2 n}{6b^2}\left(\left(1 - \frac{1}{2^b}\right)(1 - G(X)) + \frac{1}{2^b}\right)\right)
\end{aligned}
$$

where the inequality follows from the Chernoff upper bound. By a very similar calculation

$$
\begin{aligned}
\Pr\left[\hat{G} \leq G(X) - \epsilon\left(G(X) + \frac{1}{2^b - 1}\right)\right] &= \Pr\left[\frac{2^b}{2^b - 1}\bar{c} - \frac{1}{2^b - 1} \leq G(X) - \epsilon\left(G(X) + \frac{1}{2^b - 1}\right)\right] \\
&= \Pr\left[\bar{c} \leq (1 - \epsilon)\left(\left(1 - \frac{1}{2^b}\right)G(X) + \frac{1}{2^b}\right)\right] \\
&= \Pr\left[\bar{c} \leq (1 - \epsilon)\operatorname{E}[\bar{c}]\right] \\
&\leq \exp\left(-\frac{\epsilon^2 n}{6}\operatorname{E}[\bar{c}]\right) \\
&= \exp\left(-\frac{\alpha^2 \epsilon^2 n}{6b^2}\left(\left(1 - \frac{1}{2^b}\right)(1 - G(X)) + \frac{1}{2^b}\right)\right)
\end{aligned}
$$

where the inequality follows from the Chernoff lower bound. Combining the above we have

$$\Pr\left[\left|\hat{G} - G(X)\right| \geq \epsilon\left(1 - G(X) + \frac{1}{2^b - 1}\right)\right] \leq 2\exp\left(-\frac{\alpha^2 \epsilon^2 n}{6b^2}\left(\left(1 - \frac{1}{2^b}\right)(1 - G(X)) + \frac{1}{2^b}\right)\right)$$

and rearranging proves the lemma. $\qquad \square$

Now the first sample complexity bound in the theorem follows immediately from Lemma G.1. As for the second sample complexity bound, since $C(X) = -\log(1 - G(X))$ we have

$$
\left|C(X) - \hat{C}\right| = \left|\log(1 - \hat{G}) - \log(1 - G(X))\right| = \left|\log\frac{1 - \hat{G}}{1 - G(X)}\right|
$$
$$
\leq \log\left(1 + \frac{\left|\hat{G} - G(X)\right|}{1 - G(X)}\right) \leq \frac{\left|\hat{G} - G(X)\right|}{1 - G(X)}.
\tag{7}
$$

Observe that $G(X) \in [\frac{1}{k}, 1]$ for any random variable $X$ with support size $k$. Now we consider two cases. First, assume $2^b > k$. Since

$$n \geq \frac{24k^2 b^2 \log\frac{2}{\delta}}{\alpha^2 \epsilon^2 \min\{k, 2^b\}} = \frac{6b^2 \log\frac{2}{\delta}}{\alpha^2(\frac{\epsilon}{2})^2\frac{1}{k}} \geq \frac{6b^2 \log\frac{2}{\delta}}{\alpha^2(\frac{\epsilon}{2})^2\left(\left(1 - \frac{1}{2^b}\right)\frac{1}{k} + \frac{1}{2^b}\right)} \geq \frac{6b^2 \log\frac{2}{\delta}}{\alpha^2(\frac{\epsilon}{2})^2\left(\left(1 - \frac{1}{2^b}\right)(1 - G(X)) + \frac{1}{2^b}\right)}$$

we have by Lemma G.1 that

$$\left|\hat{G} - G(X)\right| \leq \frac{\epsilon}{2}\left(1 - G(X) + \frac{1}{2^b - 1}\right) \leq \epsilon(1 - G(X)),$$

where the second inequality uses the fact that $2^b > k$ implies $\frac{1}{2^b-1} \leq \frac{1}{k} \leq 1 - G(X)$. Combining with Eq.(7) we have $|\hat{C} - C(X)| \leq \frac{|\hat{G}-G(X)|}{1-G(X)} \leq \epsilon$.

Next, assume $2^b \leq k$. We have

$$n \geq \frac{24b^2k^2\log\frac{2}{\delta}}{\alpha^2\epsilon^2\min\{k, 2^b\}} = \frac{6b^2\log\frac{2}{\delta}}{\alpha^2(\frac{\epsilon}{2k})^2 2^b} \geq \frac{6b^2\log\frac{2}{\delta}}{\alpha^2(\frac{\epsilon}{2k})^2\left(\left(1 - \frac{1}{2^b}\right)(1 - G(X)) + \frac{1}{2^b}\right)},$$

where the second inequality uses $2^b \geq 1$ and $1 - G(X) \leq 1$. Thus by Lemma G.1

$$\left|\hat{G} - G(X)\right| \leq \frac{\epsilon}{2k}\left(1 - G(X) + \frac{1}{2^b - 1}\right) \leq \frac{\epsilon}{k}.$$

Combining with Eq. (7) we have

$$|\hat{C} - C(X)| \leq \frac{\left|\hat{G} - G(X)\right|}{1 - G(X)} \leq k\left|\hat{G} - G(X)\right| \leq \epsilon.$$

$\square$