# OpenReview forum: "Private and Communication-Efficient Algorithms for Entropy Estimation"
_NeurIPS.cc/2022/Conference — NeurIPS 2022 Accept_

### Official Review · Reviewer_aeFq · 2022-07-10

**Rating:** 7
**Confidence:** 3
**Soundness:** 4 excellent
**Presentation:** 3 good
**Contribution:** 3 good

**Summary:**

This paper provides a method for computing the Shannon, Gini, and collision
entropy of a distribution in the sequentially-interactive differential
privacy models.

For Shannon entropy, they approximate the weight of the Chow-Liu tree
using private l_1 estimation between pairs of variables, combined with the fact
that the l_1 estimation of a distribution can approximate entropy when the
domain is not too large. Under sequential interaction, they show that a
near-linear number of samples can be taken by estimating the weight of
a maximum spanning tree. This improves over the work of Chow and Liu.
They also show that their algorithm has optimal dimensionality dependence with a
lower bound.

They specialize these results to when the dependency graph of the distribution
is a chain or a star.

Finally, they show how to use hash collisions to estimate Gini and collision
entropy in the non-interactive local DP setting.


**Questions:**

What do the authors expect for the approximation quality of the entropy estimation algorithm given that the algorithm only considers tree-structured distributions?

Can we make any statement about sample complexity when the privacy parameter $\alpha$ is low?

**Limitations:**

The Shannon entropy of the best tree-structured distribution
is returned, this is an approximation of the true Shannon entropy which is the
assumption of the Chow-Liu tree.

(Edit: this seems to be a common assumption, so it’s not a severe limitation.)

This limitation could best be overcome with an experiments section---the
algorithms are straightforward to implement, and one could estimate the quality
of the entropy estimation on a real-example.


**Strengths And Weaknesses:**

+ The paper achieves subquadratic sample complexity for the problem of Entropy
  estimation, improving over the Chow-Liu algorithm.

+ The communication overhead is a small constant for both algorithms.

- The lower bounds do not depend on $\alpha$, the privacy parameter, thus they
  are definitely loose when \alpha is small.

- There are no experiments to evaluate the approximation quality. (Edit - they've added some which favorably compare their work)

---

> ### Author Response · Authors · 2022-08-02
> **Rebuttal**
>
> Thanks for the thorough review. We address questions and concerns below:
>
> **No evaluation of the impact of the tree-structured assumption on estimation error.**
>
> Note that the tree-structured assumption cannot be lifted without making the problem intractable. This is because a distribution on d variables has support size $\Omega(2^d)$, and so existing lower bounds [1] imply that at least $\Omega((2^d) / d)$ samples are needed to estimate the entropy of the distribution, unless we make further assumptions.
>
> Of all the assumptions that make the problem tractable, the tree-structured assumption is by far the most common. The seminal paper by Chow-Liu [2] for estimating a tree-structured distribution has 3000+ citations and has been used in numerous applications in a wide variety of scientific fields. To take one recent example, it was used to win the NIST contest on synthetic data generation [3].
>
> **Lower bound does not depend on $\alpha$, the privacy parameter.**
>
> While we did not provide a citation, a lower bound of $\Omega(1 / \alpha^2)$ follows immediately from Section 4.1 of [4]. We will add this to the final version of the paper.
>
> [1] [Valiant and Valiant (2013) “Estimating the Unseen: Improved Estimators for Entropy and other Properties”](https://papers.nips.cc/paper/2013/file/53c04118df112c13a8c34b38343b9c10-Paper.pdf)
>
> [2] [Chow and Liu (1968) "Approximating discrete probability distributions with dependence trees"](https://citeseerx.ist.psu.edu/viewdoc/download?doi=10.1.1.133.9772&rep=rep1&type=pdf)
>
> [3] [McKenna et al (2021) “Winning the NIST Contest: A scalable and general approach to differentially private synthetic data”](https://arxiv.org/pdf/2108.04978.pdf)
>
> [4] [Acharya et al (2017) “Differentially Private Testing of Identity and Closeness of Discrete Distributions”](https://arxiv.org/pdf/1707.05128.pdf)

---

> > ### Comment · Reviewer_aeFq · 2022-08-04
> > **Reviewer response**
> >
> > The authors have pointed out that the Chow-Liu assumption is a standard one to make in practice. They have addressed my concern about the lower bound, though their upper bound is still much worse for small alpha than the lower bound.
> >
> > I have updated my score of the paper accordingly. Still, I think the paper could be strengthened more with an experiments section.

---

### Official Review · Reviewer_ejtY · 2022-07-11

**Rating:** 5
**Confidence:** 3
**Soundness:** 3 good
**Presentation:** 3 good
**Contribution:** 3 good

**Summary:**

The authors study the problem of estimating the entropy of a distribution in a distributed setting. Specifically, samples from a distribution are distributed across machines. The goal is for a central entity to estimate the entropy of the distribution. They consider a number of different communication models - (a) non-interactive model, in which all users simultaneously exchange information with the server during a single round of communication (b) sequentially interactive model, in which the server queries users one at a time, possibly in an adaptive manner. They use the blackboard model of communication where the information sent from a machine to the server is broadcast to all machines for free, i.e. the information is written on a metaphorical blackboard and all machines get to view the blackboard for free.

Note that entropy estimation can be very trivially accomplished by estimating the distribution itself, but here they ensure that their algorithms are "private" where privacy is defined according to the notion of "local differential privacy".

Their main contributions are: (1) An interactive and private algorithm for estimating the Shannon entropy (2) non-interactive algorithms for estimating the Gini and collision entropy.

My reviewing capacity: I have some previous experience in doing research in distributed statistical estimation. I am comfortable with basic information theory, randomized algorithms, concentration inequalities, basic analysis. I have taken a look at the proofs in moderate detail.



**Questions:**

See suggestions for improvement above.

**Limitations:**

See comments above.

**Strengths And Weaknesses:**

The problem of estimating entropy has been well studied, and this has been studied in distributed settings as well. The paper provides improved algorithms in the settings considered. Other than for Shannon entropy, the authors provide non-interactive algorithms. Obviously it is "better" to provide strong non-interactive algorithms because non-interactive algorithms are trivially interactive with just zero interaction. While related work is well cited, more discussion about the importance of the problem is required. The algorithmic techniques are non-obvious. The analysis uses quite basic information theory and concentration inequalities.
As I said, I haven't perused the proofs in complete details, but the paper seems sound technically. The analysis is rigorously done. Also, the paper is very clearly written.

I am not very convinced about the strength and importance of the results. For example, with Shannon entropy, which really is the most commonly used form of entropy, they assume that the support size of the distribution is constant. If support size is constant, then distribution estimation can be done with O_eps,delta(1) amount of communication, and in turn entropy estimation can be done trivially. While I don't have context of the relevant work, the authors must clearly describe why they can't use techniques for distributed distribution estimation and why new techniques are required. Another issue is the structure assumed in the distribution. There is no motivation provided about why the distributional structure would be something of interest.
Also, I feel that the authors have provided very little motivation about why these problems are important. This is especially important with Gini and collision entropy.

---

> ### Author Response · Authors · 2022-08-02
> **Rebuttal**
>
> Thanks for the thorough review. We address questions and concerns below:
>
> **Can you provide some additional motivation for this paper?**
>
> We emphasize three motivations:
>
> 1. The original paper by Chow and Liu [1] has 3000+ citations and their algorithm has been used for numerous applications in a wide variety of scientific fields. Recent work [2, 3] has shown that the Chow-Liu algorithm is the optimal algorithm for estimating tree-structured distributions. This leads to a natural question: Is Chow-Liu the optimal algorithm for estimating an important statistic of a tree-structured distribution, such as its entropy? Our paper shows that the answer, surprisingly, is “no”. We show that the Chow-Liu algorithm requires $\Omega(d^2)$ pairwise samples to estimate Shannon entropy, while our algorithm requires only $O(d)$ pairwise samples.
>
> 2. In lines 29 - 34 we described wearable health-monitoring devices as a natural application for private and distributed entropy estimation. Another application is fingerprinting detection on the web [4]. Many websites track users without their consent by recording information about their devices, a practice called fingerprinting. Entropy is the standard metric used to quantify the identifiability of the collected fingerprints. So a private and distributed method for estimating entropy can be used by a browser to warn users that this covert tracking is occurring, without ever storing the fingerprints themselves. This approach has been proposed by Google Chrome [5]
>
> 3. In NeurIPS 2021, the authors of [6] posed an open problem: Is there a non-interactive private algorithm for estimating Gini entropy with sample complexity that has no dependence on the support size of the distribution? (See page 10 of their paper, final paragraph.) We answer this question affirmatively, thereby resolving the open problem.
>
> **If support size is constant, then distribution estimation can be done with O_eps,delta(1) amount of communication, and in turn entropy estimation can be done trivially.**
>
> There seems to be a misunderstanding. While the support size of each variable is constant, there are $d$ variables, so the support size of the distribution is at least $\Omega(2^d)$, which is exponential in the number of variables.
>
> **Clearly describe why they can't use techniques for distributed distribution estimation and why new techniques are required. Note that entropy estimation can be very trivially accomplished by estimating the distribution itself.**
>
> We have shown that estimating the distribution itself is much less efficient. Specifically:
>
> - The Chow-Liu algorithm is the optimal algorithm for estimating a tree-structured distribution. We show that it requires $\Omega(d^2)$ pairwise samples to estimate the entropy of the distribution, while our algorithm requires only $O(d)$ pairwise samples.
>
> - Without the tree-structured assumption, any algorithm requires $O(k / \log k)$ samples to estimate a distribution with support size $k$, while our algorithm estimates the Gini entropy of the distribution using a number of samples that is independent of support size.
>
> [1] [Chow and Liu (1968) "Approximating discrete probability distributions with dependence trees"](https://citeseerx.ist.psu.edu/viewdoc/download?doi=10.1.1.133.9772&rep=rep1&type=pdf)
>
> [2] [Daskalakis and Pan (2020) “Tree-structured Ising models can be learned efficiently”](https://arxiv.org/pdf/2010.14864.pdf)
>
> [3] [Bhattacharyya et al (2021) “Near-optimal learning of tree-structured distributions by Chow-Liu”](https://dl.acm.org/doi/pdf/10.1145/3406325.3451066)
>
> [4] [Laperdrix et al (2020) “Browser Fingerprinting: A Survey”](https://dl.acm.org/doi/10.1145/3386040)
>
> [5] https://github.com/mikewest/privacy-budget
>
> [6] [Butucea and Issartel (2021) “Locally differentially private estimation of functionals of discrete distributions”](https://proceedings.neurips.cc/paper/2021/file/cf8c9be2a4508a24ae92c9d3d379131d-Paper.pdf)

---

> > ### Comment · Reviewer_ejtY · 2022-08-08
> > **response**
> >
> > Thank you for the clarifying points. I read all the reviews/responses and I would still like to keep my score.

---

### Official Review · Reviewer_MyKL · 2022-07-12

**Rating:** 5
**Confidence:** 2
**Soundness:** 3 good
**Presentation:** 2 fair
**Contribution:** 2 fair

**Summary:**

In this paper, the authors focuses on communication-efficient algorithms for estimating the entropy of a distribution under privacy constraints. They propose private methods for Shannon, Ginni and Collision entropy that are the most widely used entropy types. The proposed algorithms are implemented for both interactive and non-interactive models. They showed that the estimation algorithms for each entropy satisfy local differential privacy and require minimal communication.


**Questions:**

It might be useful to provide some background why Ω(d^2) pair observations to achieve O(d) error is good. Similarly, Ω(d/ε) pair observations are necessary to achieve O(εd) error is an improvement and other similar bounds? Maybe it is clear for people who are working in this field but that makes reading the paper more difficult.

**Limitations:**

Authors mentions in Checklist that there is a gap between upper and lower bounds. First of all I couldn't find the discussion in the main part of the paper and it is not discussed how the gap between the upper and lower bound can be reduced (if possible).

**Strengths And Weaknesses:**

Strengths: To the best of my knowledge, the proposed approach is novel and solves an important problem in a distributed data setting.

Weaknesses: In my opinion, the main problem is the lack of motivation. I couldn't get why private entropy estimation is important. It is not easy to follow the paper at times as a non-expert in the field. Minor issue: why don't you refer to the previously best-known algorithm in line 14?  That could help to people to engage who are working on similar problems and knows the literature well.

---

> ### Author Response · Authors · 2022-08-02
> **Rebuttal**
>
> Thanks for the thorough review. We address questions and concerns below:
>
> **Can you provide some additional motivation for this paper?**
>
> We emphasize three motivations:
>
> 1. The original paper by Chow and Liu [1] has 3000+ citations and their algorithm has been used for numerous applications in a wide variety of scientific fields. Recent work [2, 3] has shown that the Chow-Liu algorithm is the optimal algorithm for estimating tree-structured distributions. This leads to a natural question: Is Chow-Liu the optimal algorithm for estimating an important statistic of a tree-structured distribution, such as its entropy? Our paper shows that the answer, surprisingly, is “no”. We show that the Chow-Liu algorithm requires $\Omega(d^2)$ pairwise samples to estimate Shannon entropy, while our algorithm requires only $O(d)$ pairwise samples.
>
> 2. In lines 29 - 34 we described wearable health-monitoring devices as a natural application for private and distributed entropy estimation. Another application is fingerprinting detection on the web [4]. Many websites track users without their consent by recording information about their devices, a practice called fingerprinting. Entropy is the standard metric used to quantify the identifiability of the collected fingerprints. So a private and distributed method for estimating entropy can be used by a browser to warn users that this covert tracking is occurring, without ever storing the fingerprints themselves. This approach has been proposed by Google Chrome [5]
>
> 3. In NeurIPS 2021, the authors of [6] posed an open problem: Is there a non-interactive private algorithm for estimating Gini entropy with sample complexity that has no dependence on the support size of the distribution? (See page 10 of their paper, final paragraph.) We answer this question affirmatively, thereby resolving the open problem.
>
> **Clarify the bounds for estimating Shannon entropy and the gaps between them.**
>
> It might be clearer to consider only the non-private version of our Shannon entropy estimation algorithm. As we state in lines 57 - 67, our algorithm estimates the entropy of a tree-structured distribution within $O(d)$ error using $O(d)$ pairwise samples. We also show that any algorithm that achieves $O(d)$ error requires $\Omega(d)$ pairwise samples. Therefore there is no gap between our upper and lower bounds to achieve $O(d)$ error. However, to achieve $O(\epsilon d)$ error, the gap is $O(1 / \epsilon^2)$.
>
> The previously best known algorithm is the Chow-Liu algorithm, which is non-interactive, and we show that any non-interactive algorithm requires $\Omega(d^2)$ samples. So our algorithm is better than Chow-Liu by a factor of $\Omega(d)$.
>
> **Minor issue: Why don't you refer to the previously best-known algorithm in line 14?**
>
> We felt it was nonstandard to put citations in the abstract. The previously best-known algorithm is [6], which we cite in line 71.
>
> [1] [Chow and Liu (1968) "Approximating discrete probability distributions with dependence trees"](https://citeseerx.ist.psu.edu/viewdoc/download?doi=10.1.1.133.9772&rep=rep1&type=pdf)
>
> [2] [Daskalakis and Pan (2020) “Tree-structured Ising models can be learned efficiently”](https://arxiv.org/pdf/2010.14864.pdf)
>
> [3] [Bhattacharyya et al (2021) “Near-optimal learning of tree-structured distributions by Chow-Liu”](https://dl.acm.org/doi/pdf/10.1145/3406325.3451066)
>
> [4] [Laperdrix et al (2020) “Browser Fingerprinting: A Survey”](https://dl.acm.org/doi/10.1145/3386040)
>
> [5] https://github.com/mikewest/privacy-budget
>
> [6] [Butucea and Issartel (2021) “Locally differentially private estimation of functionals of discrete distributions”](https://proceedings.neurips.cc/paper/2021/file/cf8c9be2a4508a24ae92c9d3d379131d-Paper.pdf)

---

> > ### Comment · Reviewer_MyKL · 2022-08-08
> > **Response to rebuttal**
> >
> > Thanks to the authors for their answers. I also read the other reviews and the replies to the reviews. I agree to Reviewer aeFq on the paper could be strengthened more with an experiments section and that also helps to clarify the motivation.

---

### Author Response · Authors · 2022-08-10
**Additional response, including experiments**

We thank the reviewers for reading our rebuttal and for their additional comments on our submission.

Reviewers aeFq and MyKL would like to see experiments. We fully agree that experiments can be illuminating, and we share some recent empirical results below. But we would first like to note that the papers that are most similar to ours, all of which we have improved or extended, do not contain any experiments. In particular:

- [1] and [2] proved that the Chow-Liu algorithm is optimal for estimating a tree-structured distribution. We showed that the Chow-Liu algorithm is suboptimal for estimating the entropy of a tree-structured distribution, and described an optimal algorithm.

- [3] described a sequentially interactive algorithm for privately estimating Gini entropy that observes $O(k)$ bits per sample, where $k$ is the support size of the distribution. We described a noninteractive algorithm (which is a weaker communication model) with the same sample complexity that observes only $O(1)$ bits per sample.

- [4] described a non-private algorithm for estimating collision entropy that observes $O(\log k)$ bits per sample, where $k$ is the support size of the distribution. We described a private algorithm that observes only $O(1)$ bits per sample.

In summary, it is very common in related literature to publish papers that make exclusively theoretical contributions.

Nonetheless, in response to the reviewers’ concerns, we ran experiments on our collision entropy estimation algorithm. In each trial, we drew samples from a discrete exponential distribution with support size $k = 1000$. We used the previously best-known algorithm [4], which requires $O(\log k)$ bits per sample and is not private, to estimate the collision entropy of the distribution. We also used our algorithm, which requires only 1 bit per sample and can be made differentially private. The results can be viewed in this [plot](https://imgur.com/Ol4IbG9). In case the plot is not visible: Our experiment showed that the previous algorithm has 5% estimation error after observing 10000 bits, while our algorithm has less than 3.5% estimation error. Thus our algorithm has lower error for the same communication cost, and is also more private.

The short rebuttal period did not enable us to run experiments for our other algorithms, but we expect similar results for them as well.

[1] [Daskalakis and Pan (2020) “Tree-structured Ising models can be learned efficiently”](https://dl.acm.org/doi/pdf/10.1145/3406325.3451006)

[2] [Bhattacharyya et al (2021) “Near-optimal learning of tree-structured distributions by Chow-Liu”](https://dl.acm.org/doi/pdf/10.1145/3406325.3451066)

[3] [Butucea and Issartel (2021) “Locally differentially private estimation of functionals of discrete distributions”](https://proceedings.neurips.cc/paper/2021/file/cf8c9be2a4508a24ae92c9d3d379131d-Paper.pdf)

[4] [Skorski (2016) “Evaluating Entropy for TRNGs: Fast, Robust and Provably Secure”](https://eprint.iacr.org/2016/1116.pdf)

---

### Meta-Review · Area_Chair_9SQ7 · 2022-08-23

**Recommendation:** Accept
**Confidence:** Certain

**Metareview:**

This is a solid paper on private and communication-efficient algorithms to estimate entropy in a distributed setting. The paper presents several algorithmic and theoretical results, including showing that the Chow-Liu algorithm performs much worse when only the entropy is needed and not the entire distribution. One main criticism from reviewers was the lack of motivation, which the authors clarified in the rebuttal. The authors should incorporate and highlight these clarified motivations in the paper. A key criticism from the reviewers is absence of experiments. While the authors provide some token experiments in the rebuttal (which they may include in the camera ready version), I think the paper is strong enough to stand on its theoretical footing. All of the reviewers have recommended acceptance, and I agree with this recommendation.

**Award:**

No

---

### Decision · Program_Chairs · 2022-09-14

Accept